# Research on the Range of Stiffness Variation in a 2D Biomimetic Spinal Structure Based on Tensegrity Structures

**DOI:** 10.3390/biomimetics10020084

**Published:** 2025-01-29

**Authors:** Xiaobo Zhang, Zhongcai Pei, Zhiyong Tang

**Affiliations:** School of Automation Science and Electrical Engineering, Beihang University, Beijing 100191, China; zhangxiaobo79756@buaa.edu.cn (X.Z.); peizc@buaa.edu.cn (Z.P.)

**Keywords:** variable stiffness mechanism, tensegrity, biological spinal structure, stiffness ratio, PSO algorithm

## Abstract

This paper presents a novel variable stiffness mechanism, namely the SBTDTS (Spinal Biomimetic Two-Dimensional Tensegrity Structure), which is constructed by integrating bioinspiration derived from biological spinal structures with the T-Bar mechanical design within tensegrity structures. A method for determining the torsional stiffness of the SBTDTS around a virtual rotational center is established based on parallel mechanism theory. The relationship between various structural parameters is analyzed through multiple sets of typical parameter combinations. Ultimately, the PSO (Particle Swarm Optimization) algorithm is employed to identify the optimal combination of structural parameters for maximizing the stiffness ratio, $K_{\theta \_time}$, of SBTDTS under different constraint conditions. This optimal configuration is then compared with the RAPRPM (a type of rotational parallel mechanism) under different values of μ, with an analysis of the distinct advantages of both variable stiffness structures.

## 1. Introduction

The sufficient stiffness of the constituent components is a prerequisite for robots to achieve motion positioning, control, and force output. Traditional robots, such as industrial robots, often possess components that can be considered rigid or have a constant stiffness, thereby exhibiting high positioning accuracy and enabling precise motion control. These advantages allow them to perform static or predefined tasks effectively. However, in unstructured environments, traditional rigid robots show limited performance in terms of environmental adaptability and human–robot interaction. Currently, the demand for versatility is continuously increasing in both industrial applications and research of robots. Robots equipped with variable stiffness actuators (VSAs) exhibit superior adaptability to various environments and operating conditions compared to those with constant stiffness [1]. Consequently, variable stiffness actuators have attracted significant attention from robotic researchers.

Variable stiffness actuators (VSAs) based on diverse principles, application scenarios, and characteristics have been discussed by researchers in various specialized fields [2,3,4,5,6]. These include wire-driven variable stiffness manipulators [7], VSAs based on redundantly actuated planar rotational parallel mechanisms [8], bidirectional antagonistic variable stiffness actuators [9], and adjustable lever-based variable stiffness structures [10,11]. On the other hand, addressing the current issue of stiffness adaptability can draw inspiration from the structural designs of organisms. Organisms evolved over millions of years and possess the capability to alter their stiffness within a certain range by tightening or relaxing their muscles, thereby adapting to the requirements of movement in various situations. The vertebral column is the most pivotal skeletal structure in vertebrates, serving as a critical component for maintaining static support and enabling dynamic motion. In vertebrates, the vertebral bones, in conjunction with the surrounding muscular tissues, collectively constitute a quintessential example of a bio-inspired variable stiffness structure.

This paper introduces tensegrity structures and incorporates bioinspiration derived from the articular unit structure of the spine to devise a bionic variable stiffness actuator. Tensegrity structures, as a novel class of biomimetic structures, exhibit explanatory compatibility [12] with biological structures across different scales and possess a broad spectrum of application domains, including biology [13,14,15,16], architectural and civil engineering [17,18,19,20], mechanical structures and mechanisms [21,22,23,24,25], aerospace engineering [26,27,28], materials science [29,30,31], and robotics [32,33,34]. The vertebral structure, comprising the spines and the surrounding associated muscles, can be regarded as consisting of multiple continuous parallel closed-loop mechanisms, which can be described by tensegrity structures.

## 2. Spinal Biomimetic Two-Dimensional Tensegrity Structure (SBTDTS)

### 2.1. Introduction of the Concept

Tensegrity structures represent a novel conceptual framework for mechanical systems. The term “tensegrity” is a neologism derived from the amalgamation of “tension” and “integrity”, and was first conceptualized by Fuller [35] in the 1960s. Building upon and summarizing the definitions provided by earlier researchers (Fuller [36], Emmerich [37], Snelson [38]) in the field, Pugh [39] proposed the most widely accepted definition of tensegrity: a tensegrity system is established when a set of discontinuous compression components interacts with a set of continuous tensile components to define a stable volume in space.

The biological vertebral structure constitutes an integrated system comprising bones and surrounding muscular tissues. The bones are actually suspended within a tensile network formed by the surrounding muscular tissues. Therefore, the bones can be regarded as compression components, while the muscular tissues can be considered as tensile components, together forming a tensegrity structure. Fish represent the oldest and relatively simplest vertebrates in terms of their structure, with their bodies primarily consisting of a spine that runs the entire length of their bodies, making them excellent biomimetic subjects for studying spinal structures. Taking the largemouth bass as an example, its muscular tissue along the body forms a W-shape, which can be interpreted as the coexistence of muscles both along the body’s longitudinal axis and those aligned with the fish bones, both of which have an impact on the overall structural stiffness. The relationship between muscular and bony structures in other vertebrates is broadly similar.

Based on the characteristics of this structural property, this paper proposes a two-dimensional variable stiffness tensegrity structure, namely the Spinal Biomimetic Two-Dimensional Tensegrity Structure (abbreviated as SBTDTS), as illustrated in Figure 1d. Among the various two-dimensional tensegrity structures, the T-Bar structure is one of the simplest [40], as illustrated in Figure 1c. The T-Bar structure can be infinitely extended within a two-dimensional plane. The SBTDTS (Spinal Biomimetic Two-Dimensional Tensegrity Structure) extends the top and bottom vertices of the T-Bar structure into platforms A_1_A_2_ and B_1_B_2_, respectively, allowing for repetition and serial connection in the vertical direction, thereby mimicking the composition of the entire spinal structure.

In the context of a single SBTDTS unit, it is comprised of a rotating platform A_1_AA_2_B_1_BB_2_ and a fixed part C_1_C_2_. It is important to clarify that the rods AB and C_1_C_2_ do not physically connect at the graphical intersection point. The upper rotating platform is connected to the fixed section via elastic limbs 1 and 2, while the lower platform is connected via elastic limbs 3 and 4. The elastic limbs are analogous to the muscles in a spinal structure, while the rigid components correspond to the bony segments of the spinal structure. Nodes A_1_, A_2_, B_1_, and B_2_ are equivalent to the attachment sites of muscles on bones, and the various legs represent the collective equivalents of muscles arranged in different configurations. The approximate correlation between the SBTDTS structure and the spinal structure is illustrated in Figure 1a,b,e,f.

The SBTDTS can modulate its overall stiffness by adjusting the internal forces in its elastic limbs. The biological spinal structure undergoes both translational and rotational movements during the motion of the organism, and the corresponding motion of the SBTDTS structure follows a similar pattern. The stiffness of tensegrity structures exhibits anisotropy, and the SBTDTS structure possesses distinct translational and rotational stiffnesses. When analyzing the motion of the SBTDTS, C_1_C_2_ remains fixed, while the rotating platform A_1_AA_2_B_1_BB_2_ can rotate around a virtual center of rotation O, as shown in Figure 1d. Figure 2a illustrates a simplified three-dimensional schematic of a fish robot body structure, which is constructed based on the principles of SBTDTS and comprises five structural units. The structure illustrated in Figure 2a can theoretically be infinitely extended along the Y-direction. Figure 2b depicts one of the structural units from the aforementioned structure. The composition of a structural unit is formed by the combination of a floral umbrella structure and the upper portion of the preceding floral umbrella structure. Figure 2c represents the projection of the structural unit depicted in Figure 2b onto the XY plane. When C_11_ coincides with C_12_ and C_21_ coincides with C_22_, the structure illustrated in Figure 2c becomes the SBTDTS shown in Figure 1d, indicating that the SBTDTS is a special form of this structure.

### 2.2. The Geometric Configuration of the SBTDTS Structure and Its Rotational Center

SBTDTS is a parallel mechanism, and the calculation of its stiffness conforms to the stiffness theory of parallel mechanisms. According to the method proposed by Behzadipour [41], the stiffness of the SBTDTS structure can be expressed as:(1)$$K=\sum_{i=1}^{n}\left(k_{i}-\frac{\tau_{i}}{l_{i}}\right)J_{i}^{T}J_{i}+\sum_{i=1}^{n}\frac{\tau_{i}}{l_{i}}\left[\begin{array}{cc} I & {\left[r_{i}\times \right]}^{T}\\ \left[r_{i}\times \right] & \left[r_{i}\times \right]{\left[r_{i}\times \right]}^{T} \end{array}\right]+\sum_{i=1}^{n}\tau_{i}\left[\begin{array}{cc} 0 & 0\\ 0 & \left[u_{i}\times \right]\left[r_{i}\times \right] \end{array}\right]$$

$k_{i}$,$\tau_{i}$ and $l_{i}$ represent the stiffness, elastic force, and length of limb *i*, respectively, with *i* ranging from 1 to 4.

The Jacobian matrix J serves to map the end-point coordinates p of a limb onto the coordinates l defined by the lengths of the limbs.(2)$$J=\frac{dl}{dp}={\left[\begin{array}{ccc} u_{1} & \cdots & u_{n}\\ \left[r_{1}\times \right]u_{1} & \cdots & \left[r_{n}\times \right]u_{n} \end{array}\right]}^{T}$$

For $J_{i}$ of limb *i*, it is defined as follows:(3)$$J_{i}={\left[u_{i}^{T}\;{\left(\left[r_{i}\times \right]u_{i}\right)}^{T}\right]}^{T}$$

$u_{i}$ is the unit vector of limb *I*, $r_{i}$ is the vector from the upper joint of limb *i* to the virtual rotational center. $\left[r_{i}\times \right]$ is the cross-product operator in a matrix.(4)$$\left[r_{i}\times \right]=\left[\begin{array}{ccc} 0 & -r_{iz} & r_{iy}\\ r_{iz} & 0 & -r_{ix}\\ -r_{iy} & r_{ix} & 0 \end{array}\right]$$

Regarding the geometric information of the SBTDTS, the radius of the upper mobile platform is denoted as $r_{a}$, the radius of the lower platform is denoted as $r_{c}$, and the radius of the base is denoted as $r_{b}$. *H* represents the overall height of the SBTDTS, whereas $H_{0}$ denotes the distance between the lower platform and the base. h signifies the distance from the upper part of the mobile platform to the virtual rotational center. $l_{u}$ signifies the length of limb 1 and limb 2 when the platform is in a non-rotating state, whereas $l_{m}$ represents the length of limb 3 and limb 4 in the same state. It is assumed that the elastic properties of each limb are similar to linear elastic springs. When the SBTDTS is in its neutral position, the internal forces in limb 1 and limb 2 are denoted as $\tau_{u}$, and the stiffness coefficients of these limbs are $k_{u}$. The internal forces of limb 3 and limb 4 are denoted as $\tau_{m}$, and their stiffness coefficients are denoted as $k_{m}$.(5)$$l_{u}=\sqrt{{\left(r_{b}-r_{a}\right)}^{2}+{\left(H-H_{0}\right)}^{2}}$$(6)$$l_{m}=\sqrt{{\left(r_{b}-r_{c}\right)}^{2}+{H_{0}}^{2}}$$

In the context of mechanisms with rigid linkages, such as articulated mechanisms, there exists a fixed rotational center. When multiple SBTDTS structures are interconnected to form a spine-mimicking configuration, a virtual rotational center emerges at each joint during every instant of motion. During mechanical analysis, the virtual rotational center of the SBTDTS can be considered equivalent to that of structures possessing a fixed rotational center. In contrast, the conventional rotational structure exhibits friction at its fixed rotational center, whereas the SBTDTS does not suffer from this issue. Given that this paper solely focuses on the two-dimensional motion of spinal structures, the Z-direction perpendicular to the plane of the paper and the resultant rotational degrees of freedom (DOF) do not exert an influence on the structures discussed herein. Consequently, the stiffness matrix, which is originally a 6 × 6 matrix in Equation (1), is reduced to a 3 × 3 matrix. It is noteworthy to mention that in the structure depicted in Figure 2b, the lower limbs serve a supportive role and can be decomposed into components acting within the XY plane and the XZ plane. The structure illustrated in Figure 2b is not fully equivalent to the SBTDTS, and, consequently, its stiffness matrix can not be reduced from a 6 × 6 matrix to a 3 × 3 matrix.(7)$$K=\left[\begin{array}{ccc} K_{x} & 0 & K_{x\theta }\\ 0 & K_{y} & 0\\ K_{x\theta } & 0 & K_{\theta } \end{array}\right]$$

Based on Equations (1)–(7), we conclude the following:(8)$$K_{x}=\frac{2{\left(r_{a}-r_{b}\right)}^{2}\left(k_{u}-\frac{\tau_{u}}{l_{u}}\right)}{l_{u}^{2}}+\frac{2{\left(r_{c}-r_{b}\right)}^{2}\left(k_{m}-\frac{\tau_{m}}{l_{m}}\right)}{l_{m}^{2}}+\frac{{2\tau }_{u}}{l_{u}}+\frac{2\tau_{m}}{l_{m}}$$(9)$$K_{y}=\frac{2{\left(H-H_{0}\right)}^{2}\left(k_{u}-\frac{\tau_{u}}{l_{u}}\right)}{{l_{u}}^{2}}+\frac{2H_{0}^{2}\left(k_{m}-\frac{\tau_{m}}{l_{m}}\right)}{l_{m}^{2}}+\frac{{2\tau }_{u}}{l_{u}}+\frac{2\tau_{m}}{l_{m}}$$(10)$$\begin{array}{cc} K_{x\theta } & =2\frac{{\left(r_{a}-r_{b}\right)}^{2}h-r_{a}\left(r_{a}-r_{b}\right)\left(H-H_{0}\right)}{{l_{u}}^{2}}\left(k_{u}-\frac{\tau_{u}}{l_{u}}\right)\\ & +2\frac{{\left(r_{c}-r_{b}\right)}^{2}\left(h-H\right)+\left(r_{c}-r_{b}\right)r_{c}H_{0}}{{l_{u}}^{2}}\left(k_{m}-\frac{\tau_{m}}{l_{m}}\right)+\frac{{2\tau }_{u}}{l_{u}}h+\frac{{2\tau }_{m}}{l_{m}}\\ & =K_{x}h-\frac{2r_{a}\left(r_{a}-r_{b}\right)\left(H-H_{0}\right)}{lu^{2}}\left(k_{u}-\frac{\tau_{u}}{l_{u}}\right)-\frac{2{\left(r_{c}-r_{b}\right)}^{2}H}{l_{m}^{2}}\left(k_{m}-\frac{\tau_{m}}{l_{m}}\right)\\ & +\frac{2\left(r_{c}-r_{b}\right)r_{c}H_{0}}{l_{m}^{2}}\left(k_{m}-\frac{\tau_{m}}{l_{m}}\right)-\frac{{2\tau }_{m}}{l_{m}}H \end{array}$$

The coupling stiffness coefficient $K_{x\theta }$ is a function of h. During the rotational motion of the SBTDTS, the rotating platform can be considered to undergo pure rotation about a virtual rotational center, which also serves as the decoupling center of the mechanism. On the other hand, at the virtual rotational center, $K_{x\theta }$ = 0. Utilizing this condition, the following can be derived:(11)$$K_{x}h=K_{x}\left(H-H_{0}\right)+\frac{2\left(r_{a}-r_{b}\right)r_{b}\left(H-H_{0}\right)}{{l_{u}}^{2}}\left(k_{u}-\frac{\tau_{u}}{l_{u}}\right)-\frac{{2\tau }_{u}}{l_{u}}\left(H-H_{0}\right)\;\;-\frac{2r_{b}\left(r_{c}-r_{b}\right)H_{0}}{{l_{u}}^{2}}\left(k_{m}-\frac{\tau_{m}}{l_{m}}\right)+\frac{2\tau_{m}}{l_{m}}H_{0}$$(12)$${\;h}^{*}=H{-H}_{0}+\frac{\frac{2\left(r_{a}-r_{b}\right)r_{b}\left(H-H_{0}\right)}{{l_{u}}^{2}}\left(k_{u}-\frac{\tau_{u}}{l_{u}}\right)-\frac{{2\tau }_{u}}{l_{u}}\left(H-H_{0}\right)}{K_{x}}\;-\frac{\frac{2r_{b}\left(r_{c}-r_{b}\right)H_{0}}{{l_{m}}^{2}}\left(k_{m}-\frac{\tau_{m}}{l_{m}}\right)+\frac{2\tau_{m}}{l_{m}}H_{0}}{K_{x}}$$

In its neutral posture, the pretensions acting on the upper and lower ends of the rotational platform of the SBTDTS balance each other out in the Y-direction. Hence, we obtain the following:(13)$$2\tau_{u}sin\alpha_{1}-2\tau_{m}sin\alpha_{2}=0$$

Based on geometric relationships, we have $sin\alpha_{1}=\left(H-H_{0}\right)/l_{u}$, $sin\alpha_{2}=H_{0}/l_{m}$. Therefore, the relationship between $\tau_{m}$ and $\tau_{u}$ is as follows:(14)$$\tau_{m}=\tau_{u}\frac{\left(H{-H}_{0}\right)l_{m}}{l_{u}H_{0}}$$

By substituting $\tau_{m}$ in the expression for $h^{*}$ with its corresponding value derived from the relationship Equation (14), we then have the following:(15)$$h^{*}=H-H_{0}+\frac{\frac{2\left(r_{a}-r_{b}\right)r_{b}\left(H-H_{0}\right)}{{l_{u}}^{2}}\left(k_{u}-\frac{\tau_{u}}{l_{u}}\right)-\frac{2r_{b}\left(r_{c}-r_{b}\right)H_{0}}{{l_{m}}^{2}}\left(k_{m}-\frac{\tau_{u}\left(H{-H}_{0}\right)}{l_{u}H_{0}}\right)}{K_{x}}$$

The virtual rotational center undergoes a shift as the rotation angle changes. In reality, spinal motion is accomplished through the cooperative action of multiple spinal joints. If we postulate a total rotation of 20 degrees, with this rotation being achieved collectively by 10 joints, then each individual joint would undergo an approximate deviation of 2 degrees. In the case of small deformations, the shift of the virtual rotational center during rotation can be neglected.

### 2.3. Torsional Stiffness of SBTDTS

The torsional stiffness of the SBTDTS is $K_{\theta }$. θ represents the angle of rotation, and θ=0 represents the initial position, as shown in Figure 3. Q represents the torque applied to drive the rotational platform. The rotational stiffness is defined as follows:(16)$$K_{\theta }=\frac{\partial Q}{\partial \theta }$$

For structures with variable stiffness, their stiffness K can be decomposed into a passive stiffness $K_{p}$, which is determined by their constitutive characteristics, and active stiffness $K_{a}$, which varies due to certain factors, such as pretension.(17)$$K=K_{p}+K_{a}$$

$K_{\theta }$ can also be decomposed into a passive stiffness $K_{\theta p}$, which is determined by its inherent properties and a $K_{\theta a}$ component that is influenced by pretension, such that(18)$$K_{\theta }=K_{\theta p}+K_{\theta a}$$

In the analysis of the active stiffness $K_{\theta a}$ of the SBTDTS, it can also be divided into upper and lower components, namely(19)$${\;K}_{\theta a}=K_{\theta a\_upper}+K_{\theta a\_bottom}$$

Due to the symmetry between the upper and bottom parts of the rotating platform, the active stiffness of the bottom part can be derived through the computation of the active stiffness of the upper part. Based on the analysis of planar rotational parallel mechanisms, the torque $Q_{upper}$ of the upper platform is defined as follows:(20)$$Q_{upper}=r^{T}Ef$$

E is the two-dimensional rotational matrix and $E=\left[\begin{array}{cc} 0 & 1\\ -1 & 0 \end{array}\right]$, f is the internal force, and $f=f_{0}l_{n}$, where $f_{0}$ is the amount of the internal force, $l_{n}$ is the unit vector of the limbs, $l_{n}={\;l}_{i}/\left|l_{i}\right|$*,*$\left|l_{i}\right|$ is the length of the limbs, $l_{i}$ is the vector of the limbs and $l_{i}=r_{i}-b_{i}$. When considering only the upper part for calculation, *i* = 1, 2, $K_{\theta \_upper}$ is as follows:(21)$$K_{\theta \_upper}=\frac{\partial \left(Q_{upper}\right)}{\partial \theta }=\frac{\partial \left(r^{T}Ef\right)}{\partial \theta }=\frac{\partial r^{T}}{\partial \theta }Ef+r^{T}E\frac{\partial f}{\partial \theta }\;=f_{0}\left(\frac{\partial r^{T}}{\partial \theta }E{\;l}_{n}+r^{T}E\frac{\partial {\;l}_{n}}{\partial \theta }\right)+r^{T}E{\;l}_{n}\frac{\partial f_{0}}{\partial \theta }$$

The first term on the right side of the equation is the active stiffness $K_{\theta a\_upper}$ caused by internal forces within the mechanism. Setting $M_{upper}=\frac{\partial l}{\partial \theta }=r^{T}E{\;l}_{n}$, $K_{\theta a\_upper}$ and $K_{\theta p\_upper}$ are as follows:(22)$$K_{\theta a\_upper}=f_{0}\left(\frac{\partial r^{T}}{\partial \theta }E{\;l}_{n}+r^{T}E\frac{\partial {\;l}_{n}}{\partial \theta }\right)=f_{0}\frac{\partial M_{upper}}{\partial \theta }$$(23)$$K_{\theta p\_upper}=r^{T}E{\;l}_{n}\frac{\partial f_{0}}{\partial \theta }=\left(\frac{\partial f_{0}}{\partial l}\cdot \frac{\partial l}{\partial \theta }\right)r^{T}El_{n}$$

∂l represents the extension or retraction length of the elastic limbs. $\frac{\partial f_{0}}{\partial l}=k$, k represents the stiffness of the elastic limbs.(24)$$K_{\theta p\_upper}=k{\left(r^{T}El_{n}\right)}^{2}$$(25)$$K_{\theta p\_upper}=K_{\theta p\_upper\_1}+K_{\theta p\_upper\_2}$$

When located in the neutral position,(26)$$K_{\theta p\_upper\_1|\theta =0}=K_{\theta p\_upper\_2|\theta =0}=\frac{k_{u}{\left[\left(r_{b}-r_{a}\right)h+r_{a}\left(H-H_{0}\right)\right]}^{2}}{l_{u}^{2}}$$(27)$$K_{\theta p\_upper|\theta =0}=K_{\theta p\_upper\_1|\theta =0}+K_{\theta p\_upper\_2|\theta =0}=\frac{2k_{u}{\left[\left(r_{b}-r_{a}\right)h+r_{a}\left(H-H_{0}\right)\right]}^{2}}{l_{u}^{2}}$$

Similarly,(28)$$K_{\theta p\_bottom|\theta =0}=\frac{2k_{m}{\left[\left(r_{b}-r_{c}\right)(H-h)+r_{c}H_{0}\right]}^{2}}{l_{m}^{2}}$$

For the limbs located on the left and right sides of the rotating platform, $M_{upper}$ represents, for each, respectively:(29)$$M_{upper\_1}=-(p_{1}sin\theta +q_{1}cos\theta )⁄l_{1}$$(30)$$M_{upper\_2}=-(p_{1}sin\theta -q_{1}cos\theta )⁄l_{2}$$
where $p_{1}=\left(H-H_{0}\right)h-h^{2}-r_{a}r_{b}\;and{\;q}_{1}=(H-H_{0})r_{a}-hr_{a}+hr_{b}$. The lengths of limb 1 and limb 2 on the left and right sides of the SBTDTS are, respectively, as follows:(31)$$l_{1}=\sqrt{{l_{u}}^{2}-2q_{1}sin\theta -2p_{1}(1-cos\theta )}$$(32)$$l_{2}=\sqrt{{l_{u}}^{2}+2q_{1}sin\theta -2p_{1}(1-cos\theta )}$$

$K_{\theta a\_upper}$ of limb 1 and limb 2 are the following:(33)$$K_{\theta a\_upper\_1}=f_{1}\frac{\partial M_{upper\_1}}{\partial \theta }=f_{1}\frac{-(p_{1}cos\theta -q_{1}sin\theta )l_{1}^{2}-(p_{1}sin\theta +q_{1}cos\theta {)}^{2}}{l_{1}^{3}}$$(34)$$K_{\theta a\_upper\_2}=f_{2}\frac{\partial M_{upper\_2}}{\partial \theta }=f_{2}\frac{-(p_{1}cos\theta +q_{1}sin\theta )l_{2}^{2}-(p_{1}sin\theta -q_{1}cos\theta {)}^{2}}{l_{2}^{3}}$$

$f_{1}$ and $f_{2}$ are the amount of the internal force of limb 1 and limb 2, respectively. By substituting Equations (31) and (32) into Equation (25) at θ = 0, at a neutral position, $f_{1}={\;f}_{2}$ = $\tau_{u}$, and therefore(35)$$\begin{array}{c} K_{\theta a\_upper|\theta =0}=K_{\theta a\_upper\_1}+K_{\theta a\_upper\_2}\\ =-2\tau_{u}\frac{\left[\left(H-H_{0}\right)h-r_{a}\left(r_{b}-r_{a}\right)\right]\left[\left(H-H_{0}\right)\left(H-H_{0}-h\right)+r_{b}\left(r_{b}-r_{a}\right)\right]}{{\left[{\left(H-H_{0}\right)}^{2}+{\left(r_{b}-r_{a}\right)}^{2}\right]}^{\frac{3}{2}}} \end{array}$$

Based on symmetry, we obtain the following:(36)$$K_{\theta a\_bottom|\theta =0}=-2\tau_{m}\frac{\left[\left(H-h\right)H_{0}-r_{c}\left(r_{b}-r_{c}\right)\right]\left[H_{0}\left(H_{0}-H+h\right)+r_{b}\left(r_{b}-r_{c}\right)\right]}{{\left[H_{0}^{2}+{\left(r_{b}-r_{c}\right)}^{2}\right]}^{\frac{3}{2}}}$$(37)$$K_{\theta a|\theta =0}=K_{\theta a\_upper|\theta =0}+K_{\theta a\_bottom|\theta =0}$$

### 2.4. Stability Analysis of SBTDTS

This section conducts a stability analysis of the SBTDTS, employing methodologies that are inspired by Skelton’s stability analysis of T-bar structures [42]. The first scenario pertains to the condition where the SBTDTS is free from external forces acting in the Y-direction. At one end of the rod C_1_C_2_ (C_1_ or C_2_), it is subjected to the combined effect of preload forces $\tau_{u}$ and $\tau_{m}$, as well as the load $N_{crit}$. The force diagram at point C_1_ is illustrated in Figure 4a. The resultant force $f(r_{b})$ is given by the following:(38)$$f(r_{b})=\tau_{u}cos\alpha_{1}+\tau_{m}cos\alpha_{2}+N_{crit}$$

Assuming that the rod C_1_C_2_ is a solid rod, its Young’s modulus is denoted as $E_{b}$, mass as $m(l_{0})$, length as $l_{0}=2r_{b}$, radius as $r_{0}$, and density as $\rho_{b}$. According to Eulerian instability, the buckling load $f(l_{0})$ is given by the following:(39)$$f(l_{0})=\frac{E_{b}I\pi^{2}}{{l_{0}}^{2}}=\frac{\pi^{3}E_{b}{r_{0}}^{4}}{16{r_{b}}^{2}}$$

Given that $f(r_{b})$ = $f(l_{0})$ and using Equation (38), the corresponding critical value of $m(l_{0})$ can be obtained as follows:(40)$$m(l_{0})=\rho_{b}\pi {r_{0}}^{2}l_{0}=\frac{16\rho_{b}{r_{b}}^{2}\sqrt{\tau_{u}cos\alpha_{1}+\tau_{m}cos\alpha_{2}+N_{crit}}}{\sqrt{\pi E_{b}}}$$

When the SBTDTS is free from forces in the Y-direction, the relationship between $\tau_{u}$ and $\tau_{m}$ satisfies Equation (14). Based on this, we can further derive the value of $m(l_{0})$ as follows:(41)$$m(l_{0})=\rho_{b}\pi {r_{0}}^{2}l_{0}=\frac{16\rho_{b}{r_{b}}^{2}\sqrt{\frac{\tau_{u}(Hr_{b}-Hr_{c}+H_{0}r_{c}-r_{a}H_{0})}{l_{u}H_{0}}+N_{crit}}}{\sqrt{\pi E_{b}}}$$

The second scenario pertains to SBTDTS experiencing a force T in the Y-direction, with the application point being B. Due to the fact that rod AB is not physically connected to rod C_1_C_2_, the stability analysis of the SBTDTS under force application in the Y-direction differs from that of the class 4 T-bar structure. For rod AB, in the context of force equilibrium in the Y-direction, the following can be stated:(42)$$2\tau_{u}sin\alpha_{1}+T=2\tau_{m}sin\alpha_{2}$$

Additionally, the relationship between $\tau_{u}$ and $\tau_{m}$ changes to the following:(43)$$\tau_{m}\;=\frac{({2\tau }_{u}\left(H{-H}_{0}\right){+Tl_{u})l}_{m}}{2l_{u}H_{0}}$$

The force condition at point C_1_ remains unchanged, and the value of $m(l_{0})$ is given by the following:(44)$$m(l_{0})=\rho_{b}\pi {r_{0}}^{2}l_{0}=\frac{16\rho_{b}{r_{b}}^{2}\sqrt{\frac{\tau_{u}(Hr_{b}-Hr_{c}+H_{0}r_{c}-r_{a}H_{0})}{l_{u}H_{0}}+\frac{T(r_{b}-r_{c})}{2H_{0}}+N_{crit}}}{\sqrt{\pi E_{b}}}$$

## 3. Analysis of Stiffness Variation and Optimization of Stiffness Ratio

For variable stiffness mechanisms, the range of stiffness variation or stiffness ratio is of utmost concern to researchers. The stiffness ratio $K_{\theta \_time}$ is defined as the ratio between the maximum and minimum stiffness within the deformation range of the limb, as follows:(45)$$K_{\theta \_time}=K_{\theta \_max}/K_{\theta \_min}$$

For the SBTDTS, both the length ratios of individual components and the stiffness ratios between the upper and bottom limbs influence the stiffness variation. This chapter first analyzes several typical scenarios of component length ratios and then utilizes the PSO algorithm to find the optimal solution for the stiffness ratio. This optimal solution is subsequently compared with a classic variable stiffness mechanism.

This paper employs formula derivation for the simulation of SBTDTS, followed by the application of algorithms to identify the optimal combination of dimensional parameters. The software selected for this purpose is MATLAB. This methodology is extensively utilized in the discussion and research of parameters related to parallel mechanisms and tensegrity structures [8,42,43], serving as a prevalent approach in such analyses. In engineering practice, finite element analysis (FEA) is frequently employed. However, FEA is not applicable when analyzing the SBTDTS discussed in this paper. The FEA method is suitable for analyzing stress, strain, deformation under load, and certain kinematic issues of structural components. The analysis of torsional stiffness for SBTDTS does not fall within the category of these classical problems. Additionally, the virtual rotational center of SBTDTS varies with changes in dimensional parameters, posing certain challenges in conducting rotational analysis. On the other hand, employing the FEA method to solve the parameter optimization problem discussed in this paper would require modeling for each parameter combination, which is not cost-effective in terms of computational time and complexity. During the analysis of the issues addressed in this paper, physical experiments confront analogous challenges. Additionally, the model presented in this paper is formulated within a two-dimensional space, whereas physical experiments require the construction of a model in a three-dimensional space, as exemplified by the model in Section 2.1. This three-dimensional model cannot be fully equivalent to its two-dimensional counterpart. Physical experiments require consideration of factors such as component thickness and gravity, which complicate the problem. In contrast, the model presented in this paper is confined to a two-dimensional space and does not account for factors like component thickness, gravity, friction, or fatigue that may arise during actual use. Instead, it is a simplified model tailored to the solution objective. Therefore, the analytical method employed in this paper is relatively rational and practical for addressing the problems discussed herein.

### 3.1. Stiffness Ratio in Typical Scenarios

Set the smallest length unit as d, with $r_{b}$ fixed at a constant value of 6d. To mitigate the impact of extreme values and corner point effects, the minimum value for both $r_{c}$ and $r_{a}$ is designated as 1d, while their maximum value is set to 5d. Furthermore, the range over which a spring maintains its linear elasticity is finite, and it is generally recognized that the deformation range within which it retains its linear elastic properties constitutes between 20% and 50% of its total length. Similarly, the deformation of muscle bundles in organisms also comprises a relatively small proportion of their total length. Therefore, it is assumed here that the deformation of the upper limb constitutes no more than 10% of its total length, implying that a small deformation is considered. On the other hand, when animal muscles are in motion, the stretching direction of the agonist muscle is unidirectional, either in compression or tension. In this paper, it is assumed that the deformation value of the limb in the SBTDTS is positive. The ratio of $k_{u}$ to $k_{m}$ is defined within a range of 1 to 5, and this ratio is denoted as $p_{k}=k_{u}$/$k_{m}$.

From Equations (18), (27), (28), and (37), the following can be derived:(46)$$K_{\theta |\theta =0}=K_{\theta a\_upper|\theta =0}+K_{\theta a\_bottom|\theta =0}+K_{\theta p\_upper|\theta =0}+K_{\theta p\_bottom|\theta =0}$$

In this paper, the variation of $K_{\theta \_time}$ with respect to $p_{k}$ is discussed based on the proportional relationship between H and $r_{b}$. Three classic scenarios are discussed, specifically, when H is 24d, 12d, and 8d, which correspond to cases where the height is 2 times, 1 time, and 2/3 times the width, respectively. For each distinct value of H, various values of $H_{0}$ are provided, and within each of these scenarios, twelve unique curves are generated based on different combinations of $r_{a}$ and $r_{c}$. In each graph, the curves are distinguished by different symbols corresponding to various values of $r_{a}$, and are further differentiated by various colors according to different values of $r_{c}$.

(a)H= 24d

Figure 5a presents the SBTDTS diagram for the case where H = 24d, while Figure 5b–d depict the scenarios for $H_{0}$ = 6d, 12d, and 18d, respectively. The maximum value of the stiffness ratio $K_{\theta \_time}$ is obtained from the curve with $r_{a}$ = 3d and $r_{c}$ = 5d in Figure 5c. The curves, which are colored differently according to variations in $r_{c}$, are relatively close in Figure 5d. The curves, which are marked with different symbols based on variations in $r_{a}$, exhibit distinct slopes in Figure 5c. Among them, the curve with $r_{a}$ = 3d has the steepest slope, while the curve with $r_{a}$ = 5d has the shallowest slope. The maximum values of $K_{\theta \_time}$ in Figure 5b,d are relatively close, both being in the vicinity of 3.7. In addition, the maximum values of $K_{\theta \_time}$ in Figure 5b,d are attained on the curves corresponding to ($r_{a}$ = 1d, $r_{c}$ = 5d) and ($r_{a}$ = 5d, $r_{c}$ = 5d), respectively.

(b)H= 12d

Figure 6a illustrates the graphical representation of SBTDTS when H is set to 12d, while Figure 6b,c,d depict the scenarios where $H_{0}$ is 3d, 6d, and 9d, respectively. The maximum values of $K_{\theta \_time}$ observed in Figure 6b–d are relatively close, ranging from a minimum of 1.7 to a maximum of 2.5. The maximum values of $K_{\theta \_time}$ in Figure 6b,d are attained on the curves corresponding to ($r_{a}$ = 1d, $r_{c}$ = 5d) and ($r_{a}$ = 5d, $r_{c}$ = 5d), respectively, which follows the same pattern as observed in Figure 5. However, it is noteworthy that the maximum value of $K_{\theta \_time}$ is specifically achieved in Figure 6d.

(c)H= 8d

As shown in Figure 7a, SBTDTS illustration is depicted, for H= 8d, while Figure 7b–d represent three scenarios where $H_{0}$ is 2d, 4d, and 6d, respectively. The maximum values of $K_{\theta \_time}$ observed in Figure 7b–d exhibit significant variation, ranging from a minimum of 1.35 to a maximum of 1.95. In Figure 7d, the maximum value of $K_{\theta \_time}$ is obtained on the curve ($r_{a}$ = 5d, $r_{c}$ = 5d).

Based on the analysis of Figure 5, Figure 6 and Figure 7, the following four observations can be summarized:

1.H (length of rod AB) is most crucial for enhancing $K_{\theta \_time}$. The increase in H has the most significant impact on elevating the value of $K_{\theta \_time}$.2.Variations in $H_{0}$ influence the maximum value of $K_{\theta \_time}$, and the influence pattern differs when H and $r_{b}$ are in different proportions.3.The maximum value of $K_{\theta \_time}$ is consistently observed on either the curve with parameters ($r_{a}$ = 5d, $r_{c}$ = 5d) or ($r_{a}$ = 3d, $r_{c}$ = 5d) in different figures, but the curve with parameters ($r_{a}$ = 5d, $r_{c}$ = 5d) generally exhibits a better linear relationship.4.Under various combinations of H and $H_{0}$, the curves corresponding to the maximum values of $K_{\theta \_time}$ exhibit satisfactory linearity.

### 3.2. Stiffness Optimization Based on the PSO

#### 3.2.1. PSO Algorithm and SBTDTS’s Parameter Optimization Conditions

The PSO (Particle Swarm Optimization) algorithm is an evolutionary algorithm first proposed by Kennedy in 1995, inspired by observations of bird predation behavior. Analogous to many algorithms designed to find optimal solutions, the computational logic of this algorithm begins by establishing a set of random solutions, followed by an iterative process to search for the optimal solution [44].

The PSO algorithm seeks optimal solutions by simulating the cooperative and competitive behaviors among individuals (particles). Specifically, particles update their positions by tracking these two “extremes” at each iteration step. One is the best search position that the particle has found so far ($p_{b}$); the other is the position of the particle with the highest fitness in the entire population at present ($g_{b}$). The iterative update formula for the PSO algorithm is as follows:(47)$$\left\{\begin{array}{c} v_{i}\left(t\right)=w\times v_{i}\left(t\right)+c_{1}\times rand\left(0,1\right)\times \left(p_{b}-x_{i}\left(t\right)\right)+c_{2}\times rand\left(0,1\right)\times \left(g_{b}-x_{i}\left(t\right)\right)\\ x_{i}\left(t+1\right)=x_{i}\left(t\right)+v_{i}\left(t+1\right) \end{array}\right.$$

For the parameter optimization of SBTDTS, this paper first conducts a broad parameter sweep with large step sizes to obtain a point that is close to the actual optimal solution. Taking this point as the starting point, we then apply the PSO algorithm within its neighborhood to find the parameter value that maximizes $K_{\theta \_time}$. The constraint conditions that each parameter value should satisfy are as follows:(48)$$\mathrm{subject}\mathrm{to}\;\left\{\begin{array}{c} r_{a\_min}⩽r_{a}⩽r_{a\_max}\\ r_{c\_min}⩽r_{c}⩽r_{c\_max}\\ p_{k\_min}⩽p_{k}⩽p_{k\_max}\\ H_{min}⩽H⩽H_{max}\\ 0.2H⩽H_{0}⩽0.8H\\ 0⩽h⩽H\\ \Delta K_{a}/K_{a0}⩽\mu \end{array}\right.$$

Among them, the constraint on h arises from the consideration that, in engineering practice, the virtual rotational center should be located within the interior of the mechanism. $\Delta K_{a}$ refers to the maximum change in the value of $K_{a}$ during the rotation process. $K_{a0}$ represents the active stiffness at the neutral position where θ equals 0. $K_{\theta a\_q}$ denotes the active stiffness measured at θ = 0.1 rad. This constraint ($\Delta K_{a}/K_{a0}⩽\mu$) aims to ensure that the influence on $K_{a}$ remains minimal under conditions of slight rotational movements.

#### 3.2.2. Discussion on the Performance Characteristics of RAPRPM and SBTDTS

For the purpose of comparative validation, this paper introduces a variable stiffness mechanism for comparison. The RAPRPM, proposed by Jiang, represents a structural unit of a flexible robotic fish based on planar serial–parallel redundantly actuated mechanisms. This structure also possesses the capability of varying stiffness [8]. As illustrated in Figure 8, the RAPRPM is depicted. The corresponding value of $K_{\theta |\theta =0}$ for RAPRPM is as follows:(49)$$K_{\theta |\theta =0}=-2\tau_{u}\frac{\left[Hh-r_{a}\left(r_{b}-r_{a}\right)\right]\left[H\left(H-h\right)+r_{b}\left(r_{b}-r_{a}\right)\right]}{{\left[H^{2}+{\left(r_{b}-r_{a}\right)}^{2}\right]}^{\frac{3}{2}}}+\frac{2k_{u}{\left[\left(r_{b}-r_{a}\right)h+r_{a}H\right]}^{2}}{l_{u}^{2}}$$

Under the condition that $r_{b}$ is fixed at 6d, the constraints for $r_{a}$ and $r_{c}$ are defined as follows: $r_{a\_max}$ and $r_{c\_max}$ are specified as 5d, while $r_{a\_min}$ and $r_{c\_min}$ are set to d. Additionally, $p_{k\_min}$ is set to 1, and $p_{k\_max}$ is set to 5. A comparison of stiffness multiples and linearity of stiffness variation between RAPRPM and SBTDTS is considered under the same parametric constraints. Li [43] presented the optimal parameter combinations for RAPRPM when μ is set to 0.1 and 0.2, respectively. Based on the parameters provided by Li, as shown in Table 1 and Figure 9, this paper calculates the corresponding values of $K_{\theta \_time}$ for the deformation of the limb (the deformation of the limb is compression) ranging from 0 to 10%.

When μ is set to 0.1, H/$r_{b}$ of RAPRPM is 2.215. Since SBTDTS has two structural parts, the corresponding H/$r_{b}$ should be double that of RAPRPM. Therefore, when $r_{b}$ is set to 6d, $H_{max}$ of SBTDTS is 26.58d. Similarly, when μ is set to 0.2, the ratio H/$r_{b}$ for RAPRPM is found to be 4.5, and, correspondingly, for SBTDTS, the value of $H_{max}$ should be selected as 54d. Using an interval of 1 to systematically traverse the parameter space, the initial points for the PSO algorithm are determined when μ is set to 0.1 and 0.2, respectively, as shown in Table 2.

The parameter optimization range for the PSO algorithm is set in Table 3.

For the PSO algorithm, the parameter settings are as follows: $w_{max}$ is set to 0.7, $w_{min}$ is set to 0.4, $c_{1}$ and $c_{2}$ are both set to 1.5, and $v_{max}$ is set to 0.05. The number of particles is 200. The algorithm employed in this paper adopts a strategy where the value of w decreases linearly as the number of iterations increases. The PSO algorithm is not a rigorous global optimum algorithm. In this paper, processing of the parameter range, constraints imposed on the velocity vector $v_{max}$, and the choice of a relatively large number of particles are all measures taken to prevent the solution process from being trapped in the vicinity of local optimal solutions, and thereby enhancing the robustness of the algorithm.

The optimal parameters and their corresponding $K_{\theta \_time}$ values for SBTDTS are presented in Table 4 and Figure 10, respectively.

From the comparison between Figure 9 and Figure 10, it can be deduced that, under two different constraints on the value of μ, the $K_{\theta \_time}$ of SBTDTS is consistently greater than that of RAPRPM. Meanwhile, the impact of variations in H on the SBTDTS structure is greater than that on the RAPRPM. Based on the corresponding stiffness formula and the curve variations depicted in Figure 9, a completely linear relationship exists between $K_{\theta |\theta =0}$ of the RAPRPM and the deformation ratio of the limbs $p_{lu}$. Additionally, there is a high degree of coincidence between $K_{\theta a\_q}$ and $K_{\theta a}$ of RAPRPM.

Conversely, as derived from the formula of the SBTDTS structure and the curve depicted in Figure 8, it can be obtained that the relationship between $K_{\theta |\theta =0}$ of SBTDTS and the deformation ratio $p_{lu}$ of the supporting legs is not entirely linear. Furthermore, the $K_{\theta p}$ of SBTDTS is also influenced by variations in $p_{lu}$, exhibiting minor changes. In Figure 10b, when $p_{lu}$ = 0.1, a noticeable separation between $K_{\theta a\_q}$ and $K_{\theta a}$ is observed, although their difference remains within an acceptable range. Therefore, on the whole, SBTDTS is considered to have achieved a significant increase in $K_{\theta \_time}$ at the expense of partial stability in angular variation and linearity.

In practical engineering scenarios, friction at a fixed rotational center can impact the value of $K_{\theta \_time}$. However, the SBTDTS, which does not have a fixed rotational center, is not affected by friction at the rotational center. This represents one of the advantages of the SBTDTS compared to traditional mechanisms with a fixed rotational center. It does not imply that the SBTDTS is entirely free from any frictional influences during its rotational motion. Therefore, the advantage of SBTDTS over RAPRPM in terms of $K_{\theta \_time}$ is even more significant in practical engineering applications. Based on these comparisons, it can be deduced that RAPRPM is adequate for work environments with specific stiffness variation requirements and relatively large rotational angle demands. However, in environments that demand significant stiffness variations, SBTDTS emerges as a clearly superior option.

## 4. Conclusions


(1)In this paper, a novel variable stiffness structure named SBTDTS is constructed by combining the morphological characteristics of biological spinal structures with the T-bar structure in two-dimensional tensegrity structures. Furthermore, based on the theory of parallel mechanisms, a method for calculating torsional stiffness centered on virtual rotational points is established and the torsional stiffness is decomposed into active stiffness and passive stiffness. The discussion of SBTDTS properties in this paper is constrained to two dimensions due to the fact that the influence of components in the XZ plane cannot be neglected for structures formed in three dimensions.(2)This paper delves into the relationship between $K_{\theta \_time}$ of SBTDTS and its structural parameters H, $H_{0}$, $r_{a}$, $r_{c}$, $p_{k}$ by analyzing the curve relationships between $K_{\theta \_time}$ and the deformation ratio $p_{k}$ across several typical parameter combinations (H/$r_{b}$ = 4, H/$r_{b}$ = 2, H/$r_{b}$ = 1). Among all the parameters, H (length of rod AB) has the most significant influence on $K_{\theta \_time}$. Additionally, among the various parameter combinations, the curve corresponding to the maximum $K_{\theta \_time}$ value often exhibits satisfactory linearity.(3)This paper employs the PSO algorithm to identify the parameter combinations that maximize the $K_{\theta \_time}$ of SBTDTS under various conditions, and compares these results with those of the RAPRPM, which has a fixed rotational center, under similar parameters. SBTDTS consistently achieves a higher $K_{\theta \_time}$ under different conditions, exhibiting a notable advantage over RAPRPM. However, the $K_{\theta \_time}$ curve of RAPRPM is a perfectly linear curve and the impact of the rotational angle on the active stiffness $K_{\theta a}$ of RAPRPM is relatively negligible. Therefore, SBTDTS and RAPRPM are each suitable for different application scenarios, with SBTDTS exhibiting advantages in scenarios requiring a wider range of stiffness variations.


## Figures and Tables

**Figure 1 biomimetics-10-00084-f001:**
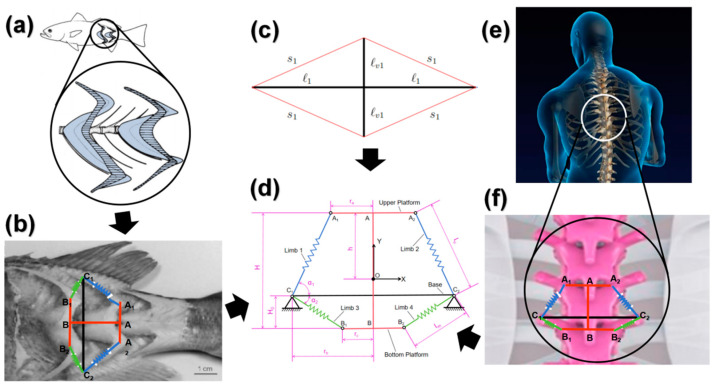
Spinal biomimetic two-dimensional tensegrity structure: (**a**,**b**) an explanation of the SBTDTS based on fish bodies, (**c**) T-Bar structure [40], (**d**) SBTDTS (rods AB and C_1_C_2_ do not physically connect at the graphical intersection point), (**e**,**f**) an explanation of the SBTDTS based on human spine.

**Figure 2 biomimetics-10-00084-f002:**
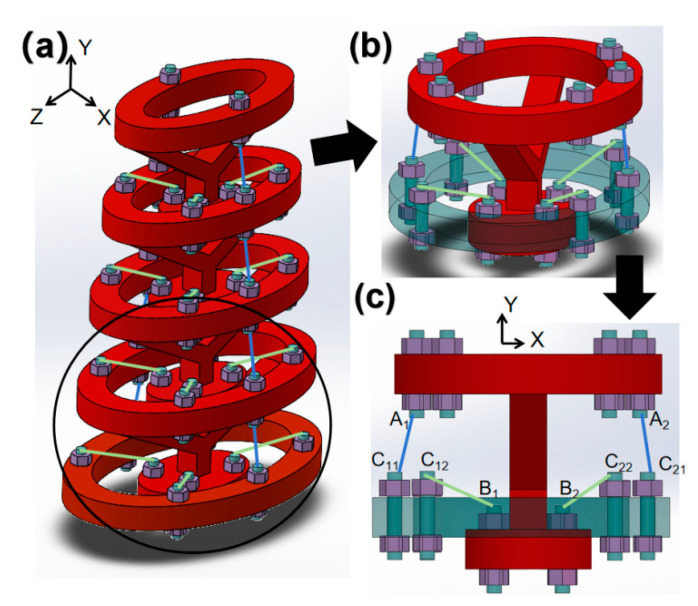
An application of SBTDTS. (**a**) Fish robot body structure composed of multiple SBTDTS structural units, (**b**) structural unit (**c**), view of the structural unit on the XY plane.

**Figure 3 biomimetics-10-00084-f003:**
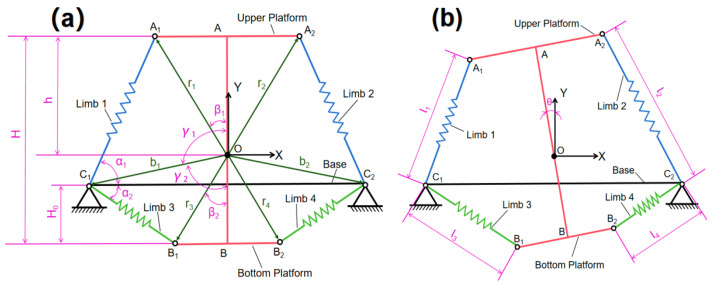
SBTDTS: (**a**) SBTDTS at neutral position, (**b**) SBTDTS rotates by a small angle.

**Figure 4 biomimetics-10-00084-f004:**
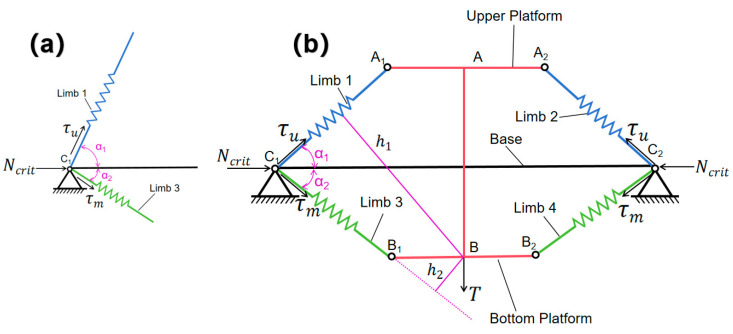
Stability analysis of SBTDTS. (**a**) No external forces act in the Y-direction. (**b**) The force acting in the Y-direction is T.

**Figure 5 biomimetics-10-00084-f005:**
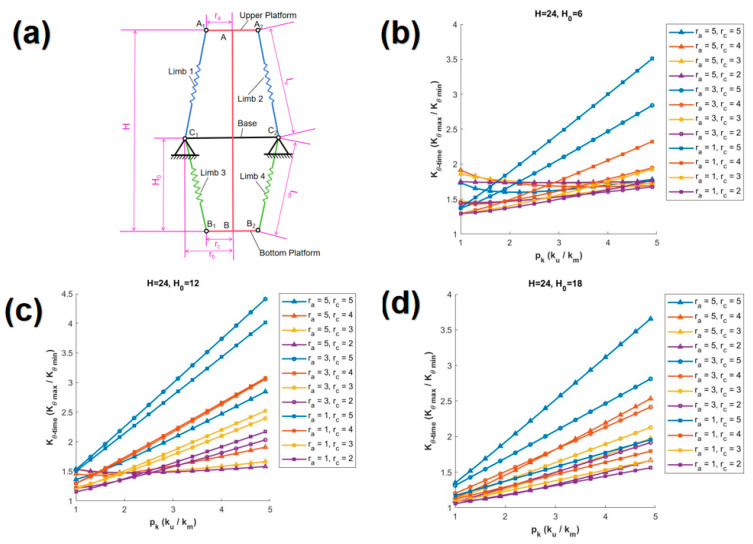
The relationship between $K_{\theta \_time}$ and $p_{k}$ (H = 24d). (**a**) Schematic diagram of the SBTDTS at H = 24d, (**b**) $H_{0}$= 6d, (**c**) $H_{0}$ = 12d, (**d**) $H_{0}$ = 18d.

**Figure 6 biomimetics-10-00084-f006:**
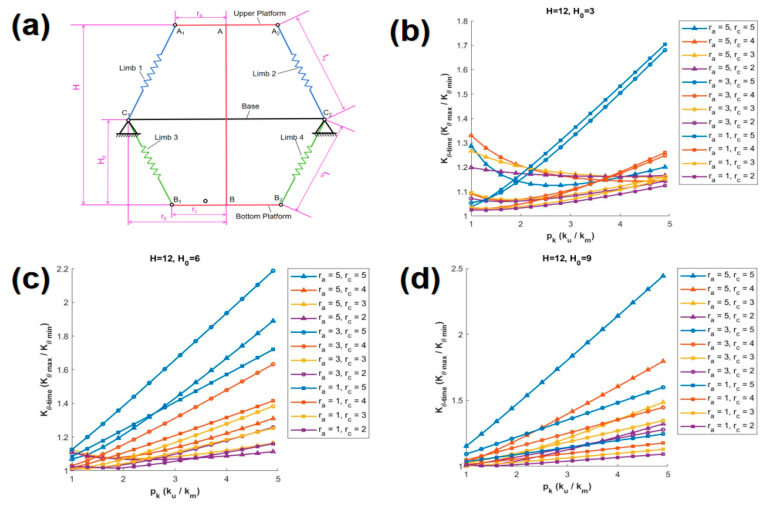
The relationship between $K_{\theta \_time}$ and $p_{k}$ (H = 12d). (**a**) Schematic diagram of the structure at H = 12d, (**b**) $H_{0}$ = 3d, (**c**) $H_{0}$ = 6d, (**d**) $H_{0}$ = 9d.

**Figure 7 biomimetics-10-00084-f007:**
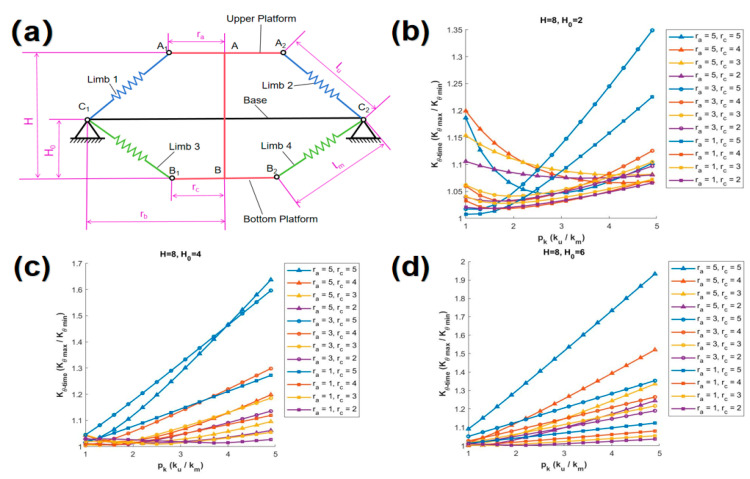
The relationship between $K_{\theta \_time}$ and $p_{k}$ (H = 8d).(**a**) Schematic diagram of the structure at H = 8d, (**b**) $H_{0}$ = 2d, (**c**) $H_{0}$ = 4d, (**d**) $H_{0}$ = 6d.

**Figure 8 biomimetics-10-00084-f008:**
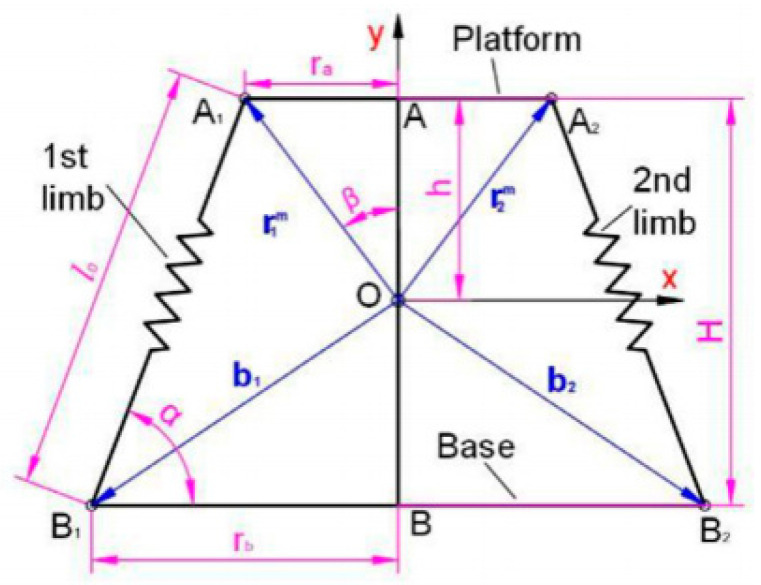
A variable stiffness mechanism—RAPRPM [8].

**Figure 9 biomimetics-10-00084-f009:**
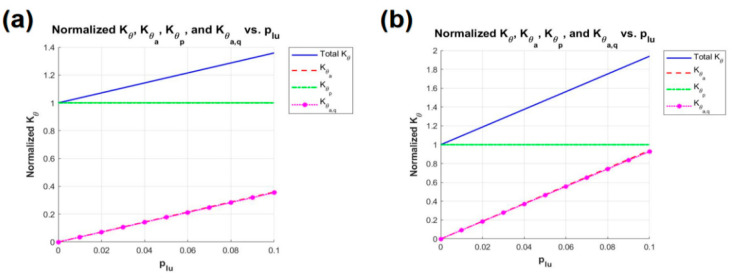
$K_{\theta \_time}$ of RAPRPM under μ is set to 0.1 and 0.2 ((**a**) μ = 0.1, (**b**) μ = 0.2).

**Figure 10 biomimetics-10-00084-f010:**
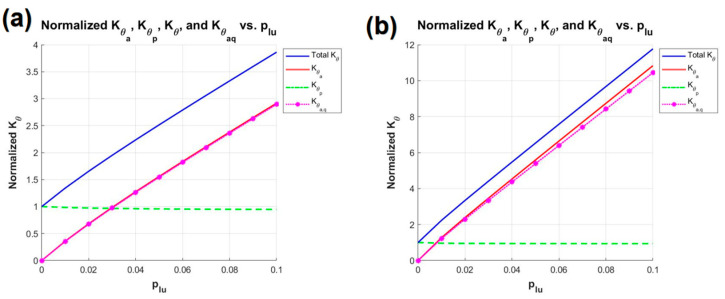
$K_{\theta \_time}$ of SBTDTS under μ is set to 0.1 and 0.2: (**a**) μ = 0.1, (**b**) μ = 0.2.

**Table 1 biomimetics-10-00084-t001:** $K_{\theta \_time}$ of RAPRPM for μ is set to 0.1 and 0.2.

μ	H	h	$r_{a}$	$r_{b}$	$K_{\theta \_time}$
0.1	288	172.5	22	130	1.359
0.2	288	149.55	30	64	1.94

**Table 2 biomimetics-10-00084-t002:** The initial points for the PSO algorithm.

μ	H	$H_{0}$	$r_{a}$	$r_{c}$	$p_{k}$
0.1	26d	12d	2d	5d	5
0.2	54d	24d	1d	5d	5

**Table 3 biomimetics-10-00084-t003:** The parameter optimization range for the PSO algorithm.

μ	H	$H_{0}$	$r_{a}$	$r_{c}$	$p_{k}$
0.1	[25d,26.58d]	[10d,14d]	[1d,3d]	[3d,5d]	[4,5]
0.2	[52d,54d]	[22d,26d]	[1d,3d]	[3d,5d]	[4,5]

**Table 4 biomimetics-10-00084-t004:** $K_{\theta \_time}$ of SBTDTS under μ is set to 0.1 and 0.2.

μ	H	$H_{0}$	$r_{a}$	$r_{c}$	$p_{k}$	$K_{\theta \_time}$
0.1	26.4245d	10.4319d	1.0023	4.2473	4.9721	3.8037
0.2	53.901d	22.2013	1.0032	4.0510	4.9388	11.3493

## Data Availability

The original contributions presented in this study are included in the article. Further inquiries can be directed to the corresponding authors.

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
