# Peer review of "Research on the Range of Stiffness Variation in a 2D Biomimetic Spinal Structure Based on Tensegrity Structures"

_biomimetics, 2025, doi:10.3390/biomimetics10020084_

Round 1
Reviewer 1 Report
Comments and Suggestions for Authors
Here’s an improved version of your text:
The research paper presents an interesting title and explores a concept that is scientifically novel. However, the lack of experimental validation undermines its impact. The mechanism design is not well-presented and remains unclear. Furthermore, according to the principles of tensegrity mechanisms, rigid components should not directly connect to one another. As presented, the design does not conform to the definition of a tensegrity mechanism.
Tensegrity structures are among the most complex mechanisms, requiring physical prototypes to validate their functionality. Simulations alone cannot fully capture the mechanical behavior or capabilities of such structures.
Suggestions for improvement:
- Develop a detailed CAD model of the mechanism to clearly illustrate its design.
- Perform FEM (Finite Element Method) analysis to evaluate its structural integrity and performance.
- Construct a prototype and test it under various conditions to validate the concept and ensure its feasibility.
Here’s an improved version of your text:
The research paper presents an interesting title and explores a concept that is scientifically novel. However, the lack of experimental validation undermines its impact. The mechanism design is not well-presented and remains unclear. Furthermore, according to the principles of tensegrity mechanisms, rigid components should not directly connect to one another. As presented, the design does not conform to the definition of a tensegrity mechanism.
Tensegrity structures are among the most complex mechanisms, requiring physical prototypes to validate their functionality. Simulations alone cannot fully capture the mechanical behavior or capabilities of such structures.
Suggestions for improvement:
- Develop a detailed CAD model of the mechanism to clearly illustrate its design.
- Perform FEM (Finite Element Method) analysis to evaluate its structural integrity and performance.
- Construct a prototype and test it under various conditions to validate the concept and ensure its feasibility.
Based on journal level, this paper is weak, can be reconsidered after major revision
Author Response
Thank you very much for reviewing my paper and providing valuable feedback.
Comments 1: The mechanism design is not well-presented and remains unclear. Furthermore, according to the principles of tensegrity mechanisms, rigid components should not directly connect to one another.
Response 1: The rod AB and rod C1C2 of this SBTDTS are not actually connected, which is consistent with your understanding of tensegrity structures. In the first edition of the manuscript, we mentioned this point in the introduction section. It is possible that our lack of emphasis on this point, coupled with its absence of annotation in the figures, has led to your misunderstanding of the structure presented in this paper. We are very sorry for this. We have emphasized this point in section 2.1 of the new manuscript and also emphasized in Figure 1 that the rods AB and C1C2 are not actually connected, and have improved the resolution of Figure 1. In addition, in order to avoid other possible misunderstandings, we have re-edited the entire text regarding the structure and related explanatory text.
Comments 2: Develop a detailed CAD model of the mechanism to clearly illustrate its design.
Response 2: Thank you for pointing that out. I agree with your suggestion. In Section 2.1 of the revised manuscript, we have added Figure 2. Figure 2 illustrates an application of SBTDTS and shows a unit within the structure in Figure 2(b). This modification makes the interpretation of the structure more complete and specific, enhancing the overall logical coherence of the paper.
Comments 3:Perform FEM (Finite Element Method) analysis to evaluate its structural integrity and performance.
Response 3: Thank you for pointing that out. Finite Element Analysis is indeed a highly effective simulation technique. However, it should be noted that Finite Element Analysis and the method employed in this paper, which involves deriving relationships and utilizing numerical simulations, each possess their unique strengths and are not inherently superior or inferior to each other. Rather, they have distinct applicable scopes. Finite Element Analysis is well-suited for analyzing stress, strain, deformation, and certain kinematic problems in structures. The model presented in this paper does not possess a fixed center of rotation, which poses certain challenges when using Finite Element Analysis to analyze its motion. On the other hand, utilizing the Finite Element Analysis method to solve the parameter optimization problem discussed in this paper necessitates the construction of a model for each parameter combination, which is not cost-effective in terms of both solution time and computational complexity. On the other hand, utilizing the Finite Element Analysis method to solve the parameter optimization problem discussed in this paper necessitates the construction of a model for each parameter combination, which is not cost-effective in terms of both solution time and computational complexity. The analytical paradigm employed in this paper is widely used in the parameter discussion and research of parallel mechanisms and tensegrity structures (references: R1, R2, R3), representing a common approach. The reasons for selecting this method are thoroughly discussed in the introductory section of Chapter 3.
Comments 4:Construct a prototype and test it under various conditions to validate the concept and ensure its feasibility.
Response 4: Thank you for pointing that out. Conducting experiments using physical prototypes is indeed a valuable suggestion. Our current work focuses on the three-dimensional Spinal Biomimetic Tensegrity Structure, employing a combined approach of simulation and experimental cross-validation. However, when it comes to the SBTDTS discussed in this paper, it is confined to a two-dimension space, which is based on our consideration of simplifying the scenario. The SBTDTS presented here is a special case of the structural unit shown in Figure 2(b). We further elucidate the differences in Sections 2.2 and the introductory part of Chapter 3. SBTDTS represents a specific case of the structural unit depicted in Figure 2(b). We further elucidate the distinctions within this context in Section 2.2 and the introductory portion of Chapter 3. In our future work, we will adopt a more comprehensive approach to validate the three-dimensional Spinal Biomimetic Tensegrity Structure, as well as its equivalence to the two-dimensional case under specific circumstances. Nevertheless, the structure presented in this paper is subject to the limitation of physical experimentation. In consideration of your concerns regarding the rationality and robustness of the structure presented herein, we have included an analysis of structural stability in Section 2.4.
R1: Chen,B.;Cui,Z.;Jiang, Producing negative active stiffness in redundantly actuated planar rotational parallel mechanisms. Mechanism and Machine Theory2018, 128,336-348.
R2: Montuori,R.;Skelton,R.E. Globally stable tensegrity compressive structures for arbitrary complexity. Composite Structures 2017, 179,682-694
R3: Kangkang,L.;Hongzhou,J.;Zuo,C.,et al.Variable stiffness design of redundantly actuated planar rotational parallel mechanisms.Chinese Journal of Aeronautics2017,30(02),818-826.
The revised manuscript is provided in the attachment.

Reviewer 2 Report
Comments and Suggestions for Authors
This work proposes a biomimetic spinal variable stiffness structure (SBTDTS) based on the tensegrity framework. Authors systematically analyzes the stiffness variation range through mathematical modeling and particle swarm optimization (PSO). The paper is logically structured and methodologically rigorous, demonstrating the structure’s potential applications in soft robotics through parameter optimization and simulation validation. Additionally, the study enhances its scientific rigor and persuasiveness by comparing the proposed structure with the RAPRPM framework. The overall quality is high, though there are some minor issues that require further clarification from the authors.
1. The authors restrict the model in a 2D plane, neglecting possible coupling effects from 3D scenario. A brief discussion on whether introducing asymmetric terms in the stiffness matrix would significantly alter the conclusions, or whether this structure is inherently limited to 2D applications would enhance clarity.
2. The authors employ the PSO algorithm to search for the optimal solution, but it is unclear whether this solution is a global or local optimum. It would be beneficial to verify the results using alternative optimization methods to ensure robustness
Author Response
I am deeply grateful for your review of my paper and the valuable insights you have provided.
Comments 1:The authors restrict the model in a 2D plane, neglecting possible coupling effects from 3D scenario. A brief discussion on whether introducing asymmetric terms in the stiffness matrix would significantly alter the conclusions, or whether this structure is inherently limited to 2D applications would enhance clarity.
Response 1: Thank you for pointing this out. I concur with your suggestion. We have supplemented additional content to address the deficiencies you mentioned. In the introductory section of Section 2.1, we have incorporated Figure 2, which is a schematic diagram depicting a structural component of the robotic fish we designed, highlighting potential future applications of SBTDTS. SBTDTS represents a specific case of the structural unit shown in Figure 2(b). We further elucidate the distinctions in the introductory sections of Section 2.2 and Chapter 3. In the analysis of three-dimensional structures, the stiffness matrix cannot be reduced from a 6x6 matrix to a 3x3 matrix. Therefore, the conclusions drawn in this paper are currently limited to two-dimensional scenarios, as supplemented in Chapter 4. However, we are also expanding our research on the Spinal Biomimetic Tensegrity Structure in three-dimensional contexts.
Comments 2:The authors employ the PSO algorithm to search for the optimal solution, but it is unclear whether this solution is a global or local optimum. It would be beneficial to verify the results using alternative optimization methods to ensure robustness.
Response 2: Thank you for highlighting this point. I agree with your suggestion. The Particle Swarm Optimization (PSO) algorithm is not strictly a global optimization algorithm in the classical sense, yet it possesses the capability to search for globally optimal solutions. To avoid converging to local optima, we have implemented the following strategies: Firstly, we preprocessed the value ranges of several parameters. Secondly, we set a relatively small upper limit for the particle velocities. Thirdly, we assigned a relatively large number of particles. These designs were all made with considerations for the robustness of the results. In the first version of the manuscript, we did not explicitly explain and elaborate on these designs, so we have revised Section 3.2.2 to clarify the robustness of our results. In fact, when processing the data, we conducted multiple repetitions of the optimization process, and the differences in the results were negligible. Since the initial particle positions are random, the proximity of the results from multiple calculations also indicates the robustness of the outcomes. However, the method of repetition, which is a straightforward yet less elegant approach frequently employed in engineering practice, was not emphasized in our paper. In fact, the preprocessing of value ranges, the setting of a relatively low upper limit for particle velocities, and the use of a relatively large number of particles are sufficient to ensure the robustness of the results.
The revised manuscript is provided in the attachment.

Reviewer 3 Report
Comments and Suggestions for Authors
The article introduces the Spinal Biomimetic Two-Dimensional Tensegrity Structure (SBTDTS), an innovative variable stiffness mechanism inspired by biological spinal structures and integrated with T-Bar mechanical design within tensegrity frameworks. By utilizing parallel mechanism theory, the study establishes a method for determining the torsional stiffness of the SBTDTS and analyzes the relationship between structural parameters through parametric studies.
The detailed analysis of various parameter combinations provides a good insights into how design variables influence torsional stiffness, enabling more informed and tailored designs.
While the study addresses stiffness and parameter optimization, it does not explicitly discuss the global stability of the structure. Ensuring that the entire SBTDTS is stable, even when individual member stability checks are satisfied, is critical for real-world applications. This is especially important for tensegrity structures, which can exhibit instability due to their interdependence and load distribution. Reference to relevant studies, such as R1, would strengthen this aspect of the research.
R1
Montuori R., Skelton R.E.
Globally stable tensegrity compressive structures for arbitrary complexity (2017) Composite Structures, 179, pp. 682 - 694, Cited 24 times. DOI: 10.1016/j.compstruct.2017.07.089
Author Response
I am deeply grateful for your review of my paper and the valuable insights you have provided.
Comments 1: While the study addresses stiffness and parameter optimization, it does not explicitly discuss the global stability of the structure. Ensuring that the entire SBTDTS is stable, even when individual member stability checks are satisfied, is critical for real-world applications. This is especially important for tensegrity structures, which can exhibit instability due to their interdependence and load distribution. Reference to relevant studies, such as R1, would strengthen this aspect of the research.
Response 1:I am deeply grateful for your review and acknowledgment of the content in this paper. I am particularly thankful for your review comments and the important literature references you provided. After making the changes based on your suggestions, I believe that the quality of this paper has been greatly improved. Indeed, stability issues constitute a crucial aspect of this type of structure.To elaborate on this issue in detail, we referred to the literature you provided and, based on the characteristics of the SBTDTS discussed in this paper, conducted an analysis of stability issues, which formed the content of section 2.4. Since the horizontal rod C1C2 and the vertical rod AB in SBTDTS are not connected, its situation differs significantly from that of the class 4 T-bar.
The revised manuscript is provided in the attachment.

Reviewer 4 Report
Comments and Suggestions for Authors
Dear authors, I attached a file with notes on text and figures.
I suggest to add some possible applications and to mention that wear and friction is not taken into account.

Author Response
Thank you very much for your review and affirmation of the content of this article.
Comments 1:Dear authors, I attached a file with notes on text and figures.
Response 1: I am profoundly thankful for your corrections to the terminology, format, and specific content of this paper. Your corrections represent the most meticulous attention to detail in paper reviews that I have ever encountered. Your professional expertise and work attitude deeply impress me. I have thoroughly addressed all the comments you made in your corrections, either by directly incorporating your suggestions or by revising based on a combination of your insights and my own understanding. After the revisions, I believe that the quality of the content in this paper has significantly improved. Additionally, I deeply apologize for any inconvenience these detailed issues may have caused you. For instance, I noticed that there is a disagreement between us regarding the description of the rotational center in section 3.2.2 concerning SBTDTS.This may be due to the imprecision in my phrasing. Here, I specifically refer to the fact that, due to the absence of a fixed rotation center, the SBTDTS is not susceptible to the friction forces that arise at the center of rotation, as seen in rotational mechanisms with a fixed rotation center. However, it is still subject to other forms of friction. In section 3.2.2, we conduct a parallel comparison between the SBTDTS and other mechanisms, which is not to claim that the SBTDTS is unaffected by friction during rotation. I have also conducted another round of formatting and textual checks on the paper to ensure the precision of these details. In future papers and work, I will pay greater attention to related issues.
Comments 2:I suggest to add some possible applications and to mention that wear and friction is not taken into account.
Response 2: Thank you for pointing this out. I agree with your suggestion. We have added several contents to address the deficiencies you mentioned. In section 2.1, we have included Figure 2, which depicts the fish robot body, an application of the SBTDTS that we are currently researching.In section 2.2, we elucidate the differences between the structural unit shown in Figure 2(b) and the SBTDTS. Additionally, the rotational process of SBTDTS revolves around a virtual rotational center, which can avoid the friction generated at rotational center. In section 2.2, we have provided supplementary explanations regarding this characteristic. Furthermore, the two-dimensional model presented in this paper represents an ideal mechanical analysis scenario, without considering the effects of gravity, friction, or fatigue during engineering practice. We have clarified this point in section 3.
The revised manuscript is provided in the attachment.

Round 2
Reviewer 1 Report
Comments and Suggestions for Authors
I am very satisfied with revision and paper looks better
Reviewer 3 Report
Comments and Suggestions for Authors
The paper can be published in its present form